# SAFE, a new therapeutic intervention for families of children with autism: study protocol for a feasibility randomised controlled trial

Rebecca McKenzie,[1] Rudi Dallos,[2] Jacqui Stedmon,[2] Helen Hancocks,[3] Patricia Jane Vickery,[3] Paul Ewings,[4] Andy Barton,[5] Tara Vassallo,[1] Craig Myhill[1]

[1]Institute of Education, University of Plymouth, Plymouth, UK
[2]Department of Clinical Psychology, University of Plymouth, Plymouth, UK
[3]Peninsula Clinical Trials Unit, University of Plymouth, Plymouth, UK
[4]Research Office, Research Design Service, Taunton, UK
[5]South West Research Design Service, Plymouth, UK

**Correspondence to**
Dr Rebecca McKenzie;
rebecca.mckenzie@plymouth.ac.uk

## ABSTRACT

**Introduction** Incidence of autistic traits, mental health problems, stress and poor coping is high among family members of children with autism. These problems are coupled with challenging behaviour among children with autism. Current treatment for these families is disjointed and costly. The need for whole family support is supported by the National Institute for Health and Care Excellence recommendations, developments regarding children's service provision, research and requests by families of children with autism. Despite evidence that family therapies can provide benefits to these families, efficacy has not been subject to a randomised controlled trial. Systemic Autism-related Family Enabling (SAFE) is a new family therapy intervention designed specifically for families of children with autism. We aim to establish the feasibility of running a fully powered randomised controlled trial to evaluate SAFE.

**Methods and analysis** Families of children with autism aged 3–16 years will be invited to participate. Consenting participants will be randomised 2:1 to either SAFE+support as usual or support as usual alone. The proposed primary outcome measure for the main trial will be the Systemic CORE 15. Participants will also complete proposed secondary outcome measures, indexing changes in child behaviour, child-parent attachment, anxiety and depression. Generic health economic outcome measures (EuroQol 5 dimensions and Child Health Utility 9 Dimensions) will also provide data on the feasibility of cost-effectiveness analysis. Questionnaires will be completed at baseline and 32 weeks post-allocation. Data on ability to identify, recruit, randomise, retain and collect data from families, acceptability of outcome measures, adherence of therapists and families to the intervention, appropriateness of resource use questionnaires and effectiveness of training will be collected for feasibility analysis. Qualitative data will also explore acceptability of SAFE and reasons for declining and withdrawing from the study.

**Ethics and dissemination** The current trial protocol received ethical approval from the South West-Exeter Research Ethics Committee (Ref: 17/SW/0192). The findings of the trial will be disseminated in collaboration with our Family Consultation Group and other partners. Findings will be shared locally, nationally and internationally through events, conferences and published papers.

### Strengths and limitations of this study

► The study addresses a gap in the available research data and will produce important feasibility information to inform a fully powered randomised controlled trial.
► The study explores the feasibility of using measures of family function and a range of mental health measures.
► Quantitative feasibility data are complemented by qualitative focus groups and interviews.
► The study explores the feasibility of economic analysis measures in a population, which includes adults and their children with developmental disorders.
► The participants are recruited from two NHS Trusts in adjacent counties in the South West of England, leading to potential bias. A future randomised controlled trial will extend to centres across the UK including Scotland and Wales.

**Trial registration number** ISCTRN83964946 (Pre-results) IRAS 213527

## INTRODUCTION

More than 1% of the UK population has a diagnosis of autism.[1] Families of children with autism present complex needs. Children with autism have impairments in social interaction, communication and behavioural flexibility.[2] Autism is widely accepted to have a genetic component and the Broad Autism Phenotype is disproportionally represented among family members.[3] Mental health problems are experienced by >70% of individuals with autism and >50% of their parents.[4 5] Individuals with autism suffer from high levels of anxiety and depression compared with those with other developmental conditions.[6–8] Families of children with autism have higher rates of depression, anxiety and social phobia than families with typically developing children, or children with other developmental disorders.[9] Parents of

children with autism are more likely to be hospitalised for mental disorders than parents of typically developing children[10]; and mothers of children with autism are reported to have higher unmet needs, more difficulties coping and lower satisfaction with service interactions than mothers of children with other disabilities.[11] Aside from these reported difficulties, families of children with autism can have positive family experiences, sense of well-being[12] and positive perceptions of their children.[13] Despite challenges, autism can be seen as enhancing family experience and some parents recognise that parenting a child with autism has added joy to their lives,[14] made them more appreciative,[15] more patient and compassionate.[16]

As families of children with autism often exhibit psychological morbidities alongside autism, costs of services to treat these problems are high.[17 18] Furthermore, untreated or unresponsive mental health problems impose societal costs making it hard for parents to interact effectively with services,[19] potentially worsening outcomes for children and exacerbating the substantial economic burden of autism.[18]

Explanations for high levels of affective disorders in these families include: stress associated with the condition of autism, genetic factors and intergenerational family dynamics. Parenting children with autism involves stresses associated with challenging behaviour, lack of Theory of Mind and atypical attachment behaviour displayed by children.[20] Parents of children with autism report that a consequent lack of psychological well-being exacerbates maladaptive behaviour in their children,[21] which is likely to result in unhelpful cycles of distress and hopelessness.

Studies exploring the medical histories of family members indicate that the onset of affective disorders may predate the birth of the child[9 22 23] suggesting that mental health difficulties cannot be wholly accounted for by stress involved in parenting. It seems, therefore, that these individuals may have been living with psychological distress for a long period of time. Depression and anxiety among family members have been tentatively linked to genetic factors independent of the Broad Autism Phenotype.[24] But few studies explore the intergenerational presence of affective disorder associated with autism.[9 23]

Previous research demonstrates that experience of trauma and abuse among women is associated with elevated risk of autism developing in their subsequent offspring.[25 26] Hence, mothers of children with autism are more likely than the general population to be coping with previous traumatic events. In addition, these families often encounter difficulties communicating needs to external agencies,[27] which may trigger existing tendencies for negative affect. Families of children with autism can experience positive family life, cope well with difficulties and enjoy good relationships with their children, but they represent a high-risk group, for whom treatment is disjointed, costly and inadequate.[28 29]

A more joined-up approach is required which focuses on autism-related need, coping with challenging behaviour and mental health difficulties by encouraging fundamental reflective functioning and improving family dynamics. The Systemic Autism-related Family Enabling (SAFE) study

should be placed in the context of the National Institute for Health and Care Excellence guidelines and recommendations[30 31] as well as developments regarding children's service provision proposed by the Munroe Report,[32 33] and the 'Future in Mind' children and young people's mental health report.[34] The SAFE study also reflects recommendations by other researchers working in the field.[35 36] Families of children with autism themselves highlight the importance of professionals working therapeutically with children and the wider family, in contrast to parents of children with conditions such as Down syndrome who tend to stress the support needs of their child within educational and community settings.[10]

SAFE is a systemic family therapy approach designed by experts to address autism-related needs including mental health difficulties and problematic behaviour. Systemic family therapy is a well-recognised, evidence-based psychotherapeutic approach,[37] which is recommended treatment for conditions such as conduct disorder, attention-deficit/hyperactivity disorder and anorexia nervosa.[38] Despite evidence that family therapy can provide benefits to children with autism and their parents,[39 40] its efficacy for treating this condition has not been subject to a randomised controlled trial. A comprehensive search of clinical trial registries revealed no ongoing trials assessing systemic family therapy as a treatment for autism and associated mental health problems. This is surprising given guidelines and recommendations for care; the successful use of family therapies for a range of conditions and reports documenting key areas of concern for the UK autism community,[41 42] which overwhelmingly show that families of children with autism want interventions which make real improvements to their daily life and sense of well-being. Consequently, the overarching aim of this study is to establish the feasibility of a definitive randomised controlled trial to evaluate SAFE therapy for families of children with autism.

## METHODS AND ANALYSIS
### Participants and recruitment
Our target population are families of children with autism, who do not have an intellectual impairment, between the ages of 3 and 16 years. SAFE is designed to have a visual, playful approach which draws from established principles of family therapy, where therapists and families work as collaborators to solve problems and effect change. SAFE activities are adaptable, family led and can be used flexibly according to the needs of the family and the age of the child. Children gain most from the intervention, however, if they can understand and communicate their responses to SAFE activities. Pilot data suggest that SAFE will be most effective and accessible for children who do not have severe symptoms or an intellectual impairment. Those children who were non-verbal and/or had severe communication difficulties found it difficult to engage with some activities. For this feasibility study, therefore, our target population is families of children with autism severity level 1 or 2 with no intellectual impairment. The authors are aware that high severity

levels may not in all cases exclude children from engaging with SAFE and that the relationship between IQ and severity is complex. These issues will be explored as part of the feasibility outcomes, namely our ability to recruit eligible families.

Future plans for SAFE include the development of a sister intervention which has extended non-verbal elements based on Intensive Interaction and is designed specifically to support families of children with autism and an intellectual impairment. This feasibility study focuses on families of children of school-age which fits with the priorities of one of our secondary sources of funding. Background research exploring diagnostic data for our proposed centres for the previous 2 years revealed no children without intellectual impairment diagnosed before the age of 3 years. This information strongly suggested that we would be unable to recruit any families with children below the age of 3 years. Consequently, we focused on the 3–16 age group.

Participants will be identified and recruited from two study research sites: University Hospitals Plymouth NHS Trust (PHNT) Child Development Centre and Cornwall Partnership NHS Foundation Trust Autism Spectrum Disorder Assessment Team (ASDAT). The pathways used to identify and recruit families will vary according to local practice, and the needs of the individual families being approached. Some families will receive a diagnosis during the SAFE recruitment period, and others will have been diagnosed up to 12 months before the SAFE study recruitment period starts.

Families with a diagnosis during the SAFE recruitment period will be approached by the diagnosing paediatrician, who will perform an initial eligibility check, invite the families to find out more and, if interested, refer the family to a member of the local SAFE study team. Families with a diagnosis before the SAFE study recruitment period will be identified as potentially eligible from clinic records by a suitably qualified member of the clinical team at each centre. Clinical staff in our centres and the surrounding areas are responsible for diagnosis of the children within our participating families. If the child is recruited from a diagnostic centre the clinical staff also assess eligibility. The severity levels of the children and their intellectual ability are assessed on the autism pathways in Plymouth and Cornwall by a multidisciplinary clinical team including educational and clinical psychologists, speech and language therapists and paediatricians. Assessment on the pathways occurs over a period of several months. This includes measures of IQ based on the Wechsler Intelligence Scale for Children (WISC-V)[43] and measures of intellectual functioning based on the British Ability Scales (BAS3)[44] as well as observations and detailed reports from the schools or nursery settings and the family.

All potential participant families will receive a participant information leaflet including an invitation to take part. All interested families will be able to speak to a member of the study team to discuss the study and have any questions answered. The participant information leaflet will contain information about the study in plain English. Parents will be asked to explain the information to younger children in a way that is appropriate for their child and suggestions for how to do this will be contained in the leaflet. A home visit will be arranged by a member of the study team for those families who express interest in participating. During the visit, a research assistant will provide the families with more detailed participant information and seek consent.

## Community pathway

Participants who have received either a new diagnosis, or a diagnosis within the last 12 months will also be approached through community groups, using a recruitment poster, invitation letter, reply slip, participant information leaflet and freepost envelope. These participants will be contacted by a member of the research team by telephone at which time they will discuss the study and answer questions. The families will also be asked to consent to providing the original National Health Service (NHS) diagnosis letter, which will be used by the research staff to determine eligibility to participate in the study, and legal guardianship at the first home visit.

## Inclusion criteria

► Family includes child with autism spectrum disorder (ASD), aged 3–16 years.
► Diagnosis of ASD, severity level 1 or 2.
► Diagnosed within 12 months of consenting to the study.
► If other diagnoses are present, ASD must be primary diagnosis.
► Family are willing to comply with study requirements.

## Exclusion criteria

► Children with ASD severity level 3.
► Children with ASD and intellectual impairment*.
► Serious concomitant illness in child or family, or other circumstances such that they are unable to comply with study requirements.
► Families who may be a risk to safety of research staff (this will be assessed by the clinical and research staff on the basis of clinical records, diagnosis letter and contact prior to the first home visit).
► Insufficient English language or capacity for parent/child to consent/assent to the study.

*Intellectual impairment will be identified by the clinical staff on the basis of pathway assessments described above including the WISC-V and the BAS3. Impairment will be deemed present on the basis of any of the following criteria:

► The child has a comorbid diagnosis of intellectual disability.
► Diagnosis specifies 'with accompanying intellectual impairment'.
► The child has been identified as requiring very substantial support (severity level 3) according to Diagnostic and Statistical Manual of Mental Disorders (DSM)-5 criteria for ASD.
► The child is being educated in a special school for children with intellectual disabilities.
► The child has an IQ of 70 or below.

## Study design

This is a randomised controlled, multicentred feasibility study including children with autism and their families (See the study schema shown in figure 1.) A total of 36 families will be recruited in four cohorts and each cohort will be randomised in a 2:1 ratio to receive support as usually employed (SUE) plus a programme of SAFE therapy, or SUE alone, for a period of 16 weeks. Advantages of 2:1 allocation include:

► Increased appeal for patients deciding whether to consent to randomisation.
► Increased ability to test training of therapists, and ability to deliver high-fidelity treatment.
► Minimal reduction in statistical power for between-groups comparisons in a full-scale evaluation.

► Increased ability to recruit required number of families within an area before randomising, which will be closer to the figure needed if and when the intervention is implemented.

Outcome assessors will be blinded to allocation. All participants will complete outcome measures at baseline and again at 32 weeks post-allocation via a face-to-face visit, hence each family will participate in the SAFE study for approximately 8 months. An embedded qualitative study will collect information about the feasibility and acceptability of the intervention and the study itself. Qualitative data will be collected at a Family Feedback Day after the 32-week post-allocation visits have been completed. The end date for the trial will be the date on which the last family completes the Family Feedback Day.

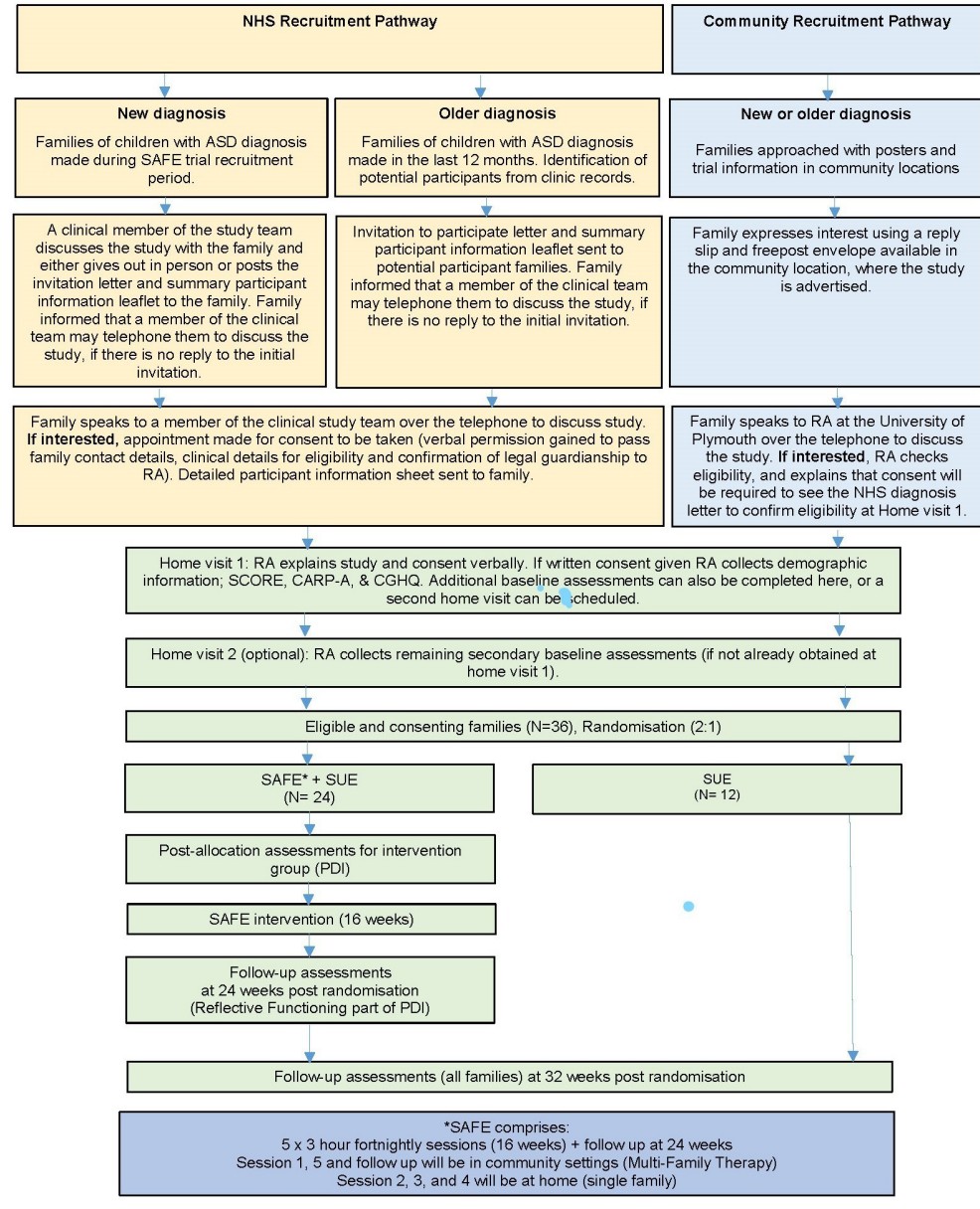

**Figure 1** Study schema. ASD, autism spectrum disorder; NHS, National Health Service; SAFE, Systemic Autism-related Family Enabling; SUE, support as usually employed.

## Outcome measures

Feasibility outcome measures:

► Ability to identify, recruit and randomise eligible families.
► Acceptability of proposed outcome measures and follow-up schedule to participants, and whether targets for loss to follow-up are achievable.
► Adherence of therapists and families to the intervention.
► Ability to gather quantitative data on outcomes.
► Appropriateness of resource use questionnaires and preference-based instruments for this population.
► Effectiveness and scalability of training arrangements.

Clinical outcome measures:

► Scores on the proposed primary outcome measure, the Systemic CORE 15 (SCORE).[45] This is a 15-item paper-based survey, which has been shown to have good internal reliability (Cronbach's α=0.89)[46] and to be a valid index of family functioning, taking approximately 20 min to complete. The SCORE is the primary measure of family functioning employed in Children and Young People's Improving Access to Psychological Therapies national programme, and is the gold standard for assessing the impact of family therapy on quality of life in the UK.[47] Every able family member will be asked to complete the SCORE, and the same family members should complete the SCORE at baseline and 32 weeks. The SCORE-15 is freely available online.[48]
► Scores on the proposed secondary outcome measures, which index changes in child behaviour, child-parent attachment, anxiety and depression.
  – Patient Health Questionnaire (PHQ)-Somatic Anxiety Depressive Symptoms. This comprises the PHQ-9 (estimated internal reliability Cronbach's α=0.86–0.89)[49] measuring depression and the generalised anxiety disorder-7 (estimated internal reliability α=0.92)[49] measuring anxiety.[50]
  – Adapted mutuality subscale of the Coding of Attachment-Related Parenting for use with children with Autism (CARP-A).[51] The CARP-A is a validated observational measure of a child with autism's attachment behaviour towards their carer. The CARP-A mutuality subscale is reported as having inter-rater reliability of 0.74.[52]
  – The Child Behaviour Checklist.[53] This is a 30-item paper-based survey, which detects emotional and behavioural problems. Reasonable internal reliability is reported for each of three scales, given that some scales only have four items: (1) competence scales (Cronbach's α=0.63–0.79), (2) problems scales (Cronbach's α=0.78–0.97) and (3) DSM orientated scales (Cronbach's α=0.72–0.91).[54]
  – The Reflective Functioning Questionnaire[53] measures ability to understand own and others' mental states (test-retest reliability coefficients are reported as 0.84).[55]
  – Caregiving Helplessness Questionnaire (CGHQ).[56] This is a 26-item questionnaire designed to assess aspects of disorganised caregiving. The CGHQ includes three subscales with reasonable internal reliability given the number of items: (1) mother helpless (α=0.86) includes seven items, (2) mother-child frightened (α=0.66) includes six items, (3) child caregiving (α=0.64) includes six items.[57 58]

► Scores on generic health economic outcome measures will also provide data on the feasibility of cost-effectiveness analysis:
  – EuroQol 5 dimensions. This is a standardised generic instrument for measuring health outcome.
  – Child Health Utility 9 Dimensions. This is a paediatric generic preference-based measure of health-related quality of life.
  – Resource Use Questionnaire (RUQ). A paper-based questionnaire completed by parent about his/her child's use of healthcare and social resources. The RUQ is designed to identify the NHS and Social Care resource use for the economic evaluation. It includes items to establish number and type of health resources being used, such as number of general practitioner (GP) visits or number of days in hospital. RUQ completion will be matched with medical records for a subgroup of families, which will help to develop strategies to minimise missing data in the future definitive trial.
► Qualitative outcomes:
  – Acceptability of SAFE and the trial process for participants and therapists.
  – Reasons for declining and withdrawing from the study.

The qualitative component will employ focus groups and interviews to investigate four key aspects of the study experience: families' experiences of the study (including intervention and potential harm of the intervention), therapists' experiences of the intervention, reasons for eligible families declining and reasons for families withdrawing from the study. After the 32-week assessments have been completed, families will be given details of the qualitative focus groups and invited to attend a family feedback day if they wish to do so. The family feedback day will involve several separate focus group sessions organised to take place over the period of a morning or an afternoon at a local venue for each centre including focus groups aimed specifically at parents and at children. The families will be told at the start of the day that they are not obliged to respond to any question or prompt if they do not wish to and that the format of the day will be open discussion with other families in response to questions presented on a screen. They will then be invited to respond to a presented topic guide exploring the four key areas stated above through discussion with each other.

## The intervention

SAFE is a manualised intensive programme of systemic family therapy designed to treat maladaptive autistic

symptoms and mental health-related difficulties encountered by families of children with autism. SAFE provides an array of therapeutic activities based on attachment theory, established systemic practice and the known visual processing preferences of people with autism.[59–61] SAFE is best seen as a toolkit with a variety of activities which can be applied to family therapy flexibly. For example, a very young child will engage with activities in a different way to teenagers. Activities include visual tasks, drawing, modelling, role-play and tracking circular patterns. Sessions are led by family need and the therapists and family work collaboratively, often in a playful way, using family resources, therapist expertise and the tools that SAFE provides. SAFE draws heavily from well-documented active and playful approaches in family therapy practice and literature.[62]

Each therapy session will include two therapists with a minimum of intermediate family therapy level of qualification and 4 days training in SAFE principles. Prior to the therapy sessions parents allocated to SAFE will complete an adapted version of the parent development interview,[60] which will provide therapists with background information on family experience. The reflective functioning questions of this interview will also be revisited as an opportunity to discuss change at the 24-week follow-up. Between weeks 1 and 16, families allocated to the SAFE intervention will attend five 3-hour SAFE therapy sessions. Sessions 1 and 5 are multifamily sessions and will take place in a community setting. Sessions 2, 3 and 4 are for individual families and will take place in a community venue or the family home. The therapists will facilitate sessions which will be video recorded, as is usual practice for therapy sessions. The videotapes will be used by the therapists in supervision sessions and preparation for subsequent sessions.

Following completion of the therapy programme, families will attend a group follow-up session at 24 weeks post-allocation. Families will discuss any changes they have encountered focusing on their ability to be reflective about challenges faced and solutions tried. Trained support workers from local voluntary groups will attend this follow-up session and will be invited to give the families information about continued support for families of children with autism through existing networks.

Each session will include the following assessments for families to complete:
1. Client Satisfaction Questionnaire 8.
2. The Helpful Aspects of Therapy Questionnaire.
3. A Between-Session Activity (BSA) homework activity. Families will be encouraged to complete a pro forma with key elements of the intervention as prompts for families to track strengths and difficulties in response to SAFE ideas. Completion of the BSA will be recorded.

At the end of each session, the therapists will also complete a training checklist and questionnaire to monitor protocol adherence.

## Support as usually employed

Families will typically be offered a post-diagnosis follow-up appointment with the diagnosing paediatrician. Parents of children whose symptoms are not severe may be directed to local authority parenting classes. Classes such as 'Timeout for Autism' (Cornwall and the Isles of Scilly) and 'Understanding Autism' (Plymouth) focus on the features of ASD, instructional parenting techniques and issues associated with education. Psychoeducation may also be offered, with families being directed to relevant resources, for example, The National Autism Society, Gateway ASD and the NHS Child and Adolescent Mental Health Services. For families where a member is experiencing depression or anxiety, treatment varies and is not linked to autism-related care. Initial referral is often through the GP. Patients may receive cognitive behavioural therapy as part of the improved access for psychological therapies service. They may also receive medication and in extreme cases a period of inpatient hospital treatment.

## Proposed sample size

In this feasibility trial no formal statistical testing of between-group differences is planned. Sample size has been selected heuristically with the goal of i) demonstrating that participants can be recruited at a rate sufficient to run a full-scale evaluation of SAFE at a later date; ii) demonstrating that it is possible to train therapists and deliver SAFE to patients within the study treatment settings and iii) demonstrate that the data collection procedures are effective, and that the data collection is acceptable to the 36 families, and not overly burdensome.

## Data analysis

Completed paper case report forms will be checked and signed by research staff before being sent to the Peninsula Clinical Trials Unit (CTU). Original case report form pages will be posted to the CTU at agreed time points for double-data entry on to a password-protected database, with copies retained at the study site. Forms will be tracked using a web-based trial management system. Data will be analysed and presented as is appropriate for a feasibility study, in particular concentrating on descriptive analyses and undertaking no formal comparisons between groups. Reporting will follow the principles of the CONSORT statement using the checklist and flow chart as recommended in the CONSORT extension for randomised pilot and feasibility.[63] The flow chart will provide details about the number of families approached, number eligible, number consenting, number randomised, number receiving allocated intervention and number assessed for outcome data at each time point. As appropriate, details will be given for individual members of the family, for example, how many family members there are and how many completed each questionnaire. Wherever possible, detailed reasons will be given for exclusions, loss to follow-up, non-completion of outcome measures, etc.

Numbers will also be provided by centre and group, to inform the logistics of recruiting nine families prior to randomisation and following them up after randomisation. For those randomised to the SAFE intervention, adherence will be reported according to the number of group sessions attended and participation of individual family members at each of the therapy sessions. Completeness of data will be reported for each outcome measure at each relevant time point. Again this will be reported for individual family members as appropriate.

For each outcome measure, the relevant scores will be calculated and presented descriptively by trial arm. Where available, published guidelines will be used to process, score and summarise the measures including, for example, the use of imputation in the event of missing items on a questionnaire. Summary measures will be calculated as appropriate, for example, means and SD, medians and ranges, numbers and percentages in categories. These measures will be presented both for baseline and for the final follow-up. The only analysis contrasting the two groups will be an interval estimate in the form of a 95% CI for the primary outcome, so that the plausibility for the effect size used in the sample size calculation for the full trial can be assessed. For this purpose, the baseline values will be used in an analysis of covariance, with acknowledgement that no effects are included for group or therapist.

Focus group interviews will be audio recorded and transcribed verbatim. Consequent qualitative data will be managed using proprietary computer-assisted qualitative data analysis software, for example, NVivo 10, and analysed thematically.[64 65] Rigour of analysis will include 'respondent validation', whereby participants are provided with a summary of their transcript and analysis so that they can assess whether the interpretations being made about the data, accurately represent them. In addition, a second qualitative researcher will conduct an independent analysis of a subset of half of the focus group transcripts. Researchers will then meet to discuss and agree the findings, which will then be presented to the Family Consultation Group for discussion.

## Patient and public involvement

Families of children with autism initiated the development of this project by communicating their complex needs and dissatisfaction with current service provision through the Plymouth Autism Network. This Network was set up by the Chief Investigator in 2011 to bring clinicians, carers, academics and individuals with autism together to share ideas, research findings and experiences. We further explored the challenges facing families of children with autism by conducting in-depth interviews and surveying over 90 families regarding their needs and the treatment they received post-diagnosis. Less than 9% of families agreed that current treatment helped with the problems they face. Our survey revealed a strong need for interventions, which support the whole family.

This pilot data led to the development of a research team within the Welcome Research Hub at Plymouth University, which included a Family Consultation Group. Our Family Consultation Group worked with us to develop and refine the SAFE intervention prior to the current project. These families have also contributed to the creation of a recruitment and treatment plan, which will be manageable for families. They have offered advice about how it is best to communicate with families at the start of the study and as it progresses. In addition, the Family Consultation Group representative is a co-applicant on this study.

Our Family Consultation Group will continue to be essential members of the team and work as an advisory group throughout the feasibility study and beyond. Formal structures are in place to ensure ongoing collaboration with the Family Consultation Group. Specifically, the representative for the Consultation Group is paid as a research assistant on the trial and is a co-applicant. She attends and actively contributes to monthly trial management group (TMG) meetings, all training sessions and fortnightly research team meetings. The representative reports key issues and requests to and from the wider group. Where necessary, additional meetings are held between the Family Consultation Group as a whole and other research staff. In these instances, travel and subsistence costs are available in line with NHS England guidelines on working with our patient and public voice partners.[66]

We see our Family Consultation Group as experts in their own lives and the lives of families with similar challenges. For this reason, we feel our role is to work with them in a supportive manner as collaborators. Their contribution is valuable in the same way as other experts on the team and we aim to facilitate one another. As stated above, the Family Consultation Group have been active in contributing to the research plan. Their input is of particular value in developing recruitment procedures, designing participant information packs and providing information about potential barriers to retention. We have also worked with them to prepare and deliver a training programme for research staff and therapists. With their help we have trained recruited staff to work in a sensitive and informed manner with participants. We also value their input in interpreting and reporting data; in particular commenting on possible ways to overcome challenges for the main randomised controlled trial.

Our families can help by identifying local networks and sharing their experience with new groups. Our Family Consultation Group are proactive campaigners for change and have extensive knowledge of existing bodies such as the National Autistic Society. They can also provide a family-centred perspective on research outcomes. They are, therefore, well-placed to collaborate with us in planning next steps and disseminating findings at local and national levels.

## ETHICS AND DISSEMINATION

### Risks and safety

Families of children with autism are a potentially vulnerable group. The risks associated with participating in this study are however, considered minimal, with no adverse events anticipated in any participant. For those in the intervention group, there is a slight chance that the SAFE family therapy sessions could lead to an initial increase in family disagreements as family members learn how to change the way they solve problems and talk with one another. However, the purpose of the intervention is ultimately to equip families with skills to handle these difficulties by learning how to change the way they solve problems and improve their communication, and the SAFE family therapists will be available to provide support and will be trained to handle any emerging problems. Should any issues arise the SAFE family therapists will have access to two consultant clinical psychologists to provide further support and advice.

During the trial, the children with autism will remain in the care of the Child Development Centre or the ASDAT and will have access to usual care should any unforeseen circumstances arise. Other members of the family will also continue to be able to seek care and advice from the GP or any other specialist services they are concurrently involved with.

### Monitoring adverse events

The research team have mechanisms in place to report serious adverse events (SAE) related to mental health. SAEs related to mental health may be volunteered by the participant or discovered by the therapists, research assistants or other member of the research team during the SAFE family therapy sessions, or as a result of direct reporting (e.g. by telephone) by a family member, independent clinician or other informant. SAEs will be recorded from the time of consent until the date the participant completes the follow-up or withdraws from the study.

If the CI considers that the SAE is not, or is unlikely to be, associated with the trial, the CTU will obtain a second assessment of causality from an independent assessor. Any SAE which in the opinion of either adjudicator is possibly related to the trial will be reported to the Research Ethics Committee (REC) within 15 days of the local research team having become aware of the event. All SAEs will be followed until either stabilised if chronic conditions, or resolved.

### Dissemination

If the feasibility study meets progression criteria an important part of our dissemination plan is to raise awareness of the need for a larger multicentre trial. We will, therefore, offer targeted summaries of our findings and presentations to policy makers. The findings will also be broadly disseminated, but in a manner appropriate to a feasibility study. We plan national conference presentations and published papers to inform clinicians, academics and therapists about the possible benefits of SAFE and generate interest in the future trial. We will make use of our existing connections including the Association for Family Therapy, the National Autistic Society and the Institute of Family Therapy to reach relevant audiences. Our qualitative findings will also be published with detailed accounts of the families' reactions to SAFE and their views on its effectiveness. We will also provide forums for participating families to share their own experiences of the intervention with wider audiences through existing networks, groups and events across the UK.

Our Family Consultation Group will be integral to our dissemination plan. Their involvement will include presenting their experiences as delegates at national and international conferences, being active co-authors on published papers, leading the organising committee for a local event sharing findings with families, key local stakeholders, clinicians and other interested partners; and liaising with other bodies to raise awareness of the study findings including Autistica, The National Autistic Society and the Brandon Trust.

### Informing potential participants of possible benefits and known risks

The participant information sheets and leaflets will provide potential participants with information about the possible benefits and risks of taking part in the trial. For example, the participants will be informed that a potential risk of receiving the SAFE therapy is that the sessions may evoke difficult emotions and feelings; this could lead to family disagreement as they move towards change. The families will also be informed that benefits of the trial include the possibility of improved coping skills when faced with challenges and contribution to finding out if SAFE can progress to a national trial. Participants will be given the opportunity to discuss risks and benefits with a member of the research team prior to consenting to participate.

### Obtaining informed consent from participants

All participants will receive a leaflet and information sheet prior to consent. There are two versions of the information sheet, one for adults and one for children. In the leaflet, parents are encouraged to explain the trial to their younger children and some guidance for doing this is provided. The information sheet states that the participants have the right to withdraw at any point during the trial and that data collected from them will be confidential. All participants will have a home-visit prior to consent from a member of the research team and will be able to ask questions or go through the information verbally. Participants will have the process of the study explained to them including the estimated time they will have to wait prior to randomisation and starting the intervention if allocated to that arm.

## Data protection/confidentiality

Participants will be given a unique identification number. The data will be pseudo-anonymised in the sense that there will be an identification number on the documentation but otherwise no means of identifying the individual to which the data relates. The research team will ensure that participants' pseudo-anonymity is maintained on all documents. Data will be collected and stored in accordance with the current legal and regulatory documentation. Electronic study records will be stored in a SQL server database, stored on a restricted access, secure server maintained by Plymouth University. Data will be entered into the database via a bespoke web-based data entry system encrypted using secure sockets layer (SSL). Access to electronic data will be permission based, and at the discretion of the clinical trials unit data management team.

Anonymised paper-based study data will be stored in locked filing cabinets. Copies of study data retained at the lead study site will be securely stored for the duration of the study prior to archiving. Video data will be transported via encrypted memory sticks and will be transferred to a password-protected computer. The clinical trials unit data team will have access to study data, including identifiable data. Other members of the study team and the trials unit will have restricted access to pseudo-anonymised study data. Access will be granted to the sponsor and host institution on request, to permit study-related monitoring, audits and inspections.

## Research governance and the conduct of the trial

The trial will be conducted to protect the human rights and dignity of the participant as reflected in the Declaration of Helsinki. An important factor in protecting the participants is ongoing consultation with the SAFE Family Consultation Group. A representative of this group is a member of the research team and is involved in decision-making processes. The research team including the family consultation group are proactive in minimising discomfort and risk for participants, respecting their wishes over science and society, respecting the right to withdraw and the need for families to have access to all relevant information.

The Chief Investigator will be responsible for the overall conduct of the study, keeping it to schedule and within budget. Working closely with the CTU, she will be the focal contact for enquiries from both sites. The CTU will manage the study, liaise with sites, monitor recruitment, work with the sponsor and report to TMG meetings. The TMG will meet regularly throughout the feasibility study. A Trial Steering Committee (TSC) will have an overarching monitoring responsibility. The TSC is expected to meet three times during the study, but will be additionally convened at the chairman or Chief Investigator's request.

## Dissemination plans

If the feasibility study demonstrates successful recruitment, data collection and an ability to deliver the intervention, an important part of the dissemination plan is to raise awareness of the need for a larger multicentre trial. Targeted summaries of the findings and presentations will be disseminated to policy makers. The findings will also be broadly disseminated, but in a manner appropriate to a feasibility study. National conference presentations and published papers will be prepared to inform clinicians, academics and therapists about our feasibility results and generate interest in the future trial. Existing connections including the Association for Family Therapy, the National Autistic Society and the Institute of Family Therapy will be used to reach relevant audiences. The qualitative findings will also be published with detailed accounts of the families' reactions to SAFE and their views on its usefulness. A summary of study results in plain English will be available on the Peninsula Clinical Trials website.

## Clinical trials authorisation and ethical approval

Clinical trials authorisation is not required. The trial will be conducted in accordance with the protocol, the principles of the Declaration of Helsinki and International Conference of Harmonisation - Good Clinical Practice (ICH GCP). Any amendments of the protocol will be submitted to the sponsor, Health Research Authority and REC for approval.

## Trial sponsorship

The trial is sponsored by PHNT.

## Trial Steering Committee

The TSC will include an independent chair and two other independent members, along with the lead investigator and the other study collaborators including a parent representative. They will meet once a year.

**Acknowledgements** The authors would like to thank the SAFE Family Consultation Group for their ongoing input and expertise. The authors would like to thank James Cook, Mary Hosken, Dr Ben Whalley, Dr Antonieta Medina-Lara and Professor Julian Archer for their help in developing and conducting this study. The authors would also like to give special recognition to the dedication, hard work and sensitivity of Helen Hancocks, the SAFE trial manager for the Peninsula Clinical Trials Unit.

**Contributors** RM, JV and HH were responsible for the overall development of the protocol. RM, HH, RD, CM, PE, AB, TV and the Family Consultation Group were involved in the conception and production of the study and the development of the initial protocol. PE and AB provided methodological expertise and advice on quantitative analysis, PE provided statistical expertise. RD was the lead researcher, with the support of JS, on design of the intervention and the qualitative component. TV, with the support of the Family Consultation Group, and CM advised on design and ethics, particularly from the participant perspective. All authors made substantial contributions to drafting, revision and approval of the document.

**Funding** This work was supported by the National Institute for Health Research (NIHR) grant number: PB-PG-0815-20058.

**Competing interests** Contributors are co-applicants or employed research staff on the SAFE project, which receives funds from both NIHR and Autistica. Professor Rudi Dallos holds joint Intellectual Property rights for the SAFE intervention with The University of Plymouth.

**Patient consent for publication** Not required.

**Ethics approval** The study has appropriate Research Ethics Committee (REC) approval from the South West-Exeter Research Ethics Committee (Ref: 17/SW/0192) and approval from the Health Research Authority (HRA).

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
