## [Reviewer comments · BMJ Open]

ARTICLE DETAILS

TITLE (PROVISIONAL)	SAFE, a new therapeutic intervention for families of children with autism: study protocol for a feasibility randomised controlled trial
AUTHORS	McKenzie, Rebecca; Dallos, Rudi; Stedmon, Jacqui; Hancocks, Helen; Vickery, Patricia; Ewings, Paul; Barton, Andy; Vassallo, Tara; Myhill, Craig

VERSION 1 - REVIEW

REVIEWER	Liz Pellicano Macquarie University, Australia
REVIEW RETURNED	22-Jul-2018

GENERAL COMMENTS	This manuscript reports a study protocol for a future, feasibility randomised controlled trial for a family-level therapeutic intervention for families of children on the autism spectrum. Many families struggle following the diagnosis of their child and so it is encouraging to see researchers focus their efforts on designing and evaluating post-diagnostic support for families. I have several comments for the authors to consider – particularly with regard to clarifying various aspects of the rationale and methods. These are listed below. To begin, however, I note that the background section was overwhelmingly negative. The authors talk about the “substantial economic burden of autism”, the “palpable air of tension” in families, children’s “maladaptive behaviour and lack of empathy” to pick just a few phrases – all of which are potentially very stigmatising. This may be the reality for many families, but it is not a necessary consequence of having a child with a diagnosis of autism, with some studies describing families’ positive perceptions and experiences (see https://www.tandfonline.com/doi/abs/10.1080/09687599.2016.1216393 and https://www.ncbi.nlm.nih.gov/pmc/articles/PMC4827600/, as two examples). Indeed, I assume that developing positive family wellbeing is the goal of this particular, SAFE, intervention. If that is the case – and this should be clearer in the introduction or in the description of the intervention itself – then the authors need to demonstrate that developing positive family wellbeing is possible for families. I suggest the authors rewrite their introduction to
--

ensure that their description of family functioning is more balanced.

I also found that there were a number of claims that were unsubstantiated (i.e., no references) and the rationale for various decisions were not provided. I have described these instances below, too.

1. p. 2, abstract. The authors state that children will be aged 3-6 years but this is not what is reported in the methods. Please correct. In fact, no rationale is given for the age range selected – which is a significant oversight, since age of diagnosis might well have an effect on the family's response to therapy. The authors need to outline their rationale for the reader.

2. p. 3, para 1. Please provide a reference for the prevalence estimate. Please also provide a reference for the claim that “numbers are rising” in the UK. As far as I am aware, there is no prevalence study showing such a rise. In fact, the study of which I am aware suggests that – unlike in the US – there has been no change in prevalence, at least from 1990s through 2010:
<https://bmjopen.bmj.com/content/3/10/e003219>

3. p. 4, para 1. The authors need to be more cautious with their claims about autism. For example, they state that autistic children demonstrate a “lack of empathy”, which is not necessarily the case. Children on the autism spectrum may well show difficulties in theory of mind and perspective-taking but do not necessarily show poor empathy. Please adjust this sentence.

4. p. 4, para 1. No reference is provided for the final sentence on attachment in this paragraph – which sounds rather tenuous.

5. p. 4 para 2. More details need to be provided regarding unpublished work, which has not been peer-reviewed and which cannot be examined by the reader.

6. p. 4, para 2. The authors state that “families of children with autism are often characterised by a palpable air of tension”. Is the phrase “palpable air of attention” from the paper cited, or is this the authors' interpretation of their results? If it's the former, the authors should provide a page number. If it's the latter, I suggest that the authors cite qualitative studies that have formally analysed parents' own perceptions of family functioning to support their claim.

7. p. 5 para 2. The target population are families of children on the autism spectrum who do not have an intellectual disability. But no rationale is provided for excluding children with additional intellectual disabilities. And indeed, nothing is presented in the methods to show how they will ascertain the intellectual ability of the child – and therefore address their inclusion/exclusion criteria.

8. p. 6 para 2. The authors state that they will be excluding children with a severity level of 3. Why exactly? Do they not believe that their intervention will be effective with this group? This needs to be properly explicated in the methods.

9. p. 6, para 3. Similarly, no rationale is provided for decisions concerning the duration of the intervention and follow-up period. In particular, why is there no follow-up immediately following the intervention (16 weeks) or at the group follow up (24 weeks)?

10. p. 7, clinical outcome measures. Insufficient psychometric information was provided on all of these measures. Reliability estimates should be provided. It would also be helpful, particularly for the primary outcome measure, which will be less well known to readers, for the authors to provide some example items and

	describe the scale that parents will need to use in response to the items. 11. p. 7. In the background provided, the authors suggest that the presence of the broader autism phenotype in parents may well moderate – and potentially mediate – family/children’s functioning. Yet I was surprised that there was no questionnaire measuring the BAP included in the secondary outcome measures. Perhaps this is another reason to revisit the background, as suggested above. 12. p. 7. Resource Use Questionnaire – is this measure meant to elicit information on the family’s ‘support as usual’? This is very unclear. Will it elicit information on the many types of interventions that parents might try to help their children (e.g. complementary and alternative medicine) beyond their accessing formal support services? 13. p. 7. The qualitative methods should try to elicit information about potential harms of the intervention, as these could well be quite subtle in nature. 14. p. 9. Please provide references for “the known visual processing preferences” of autistic people. 15. p. 9. Is the Between Session homework activity mandatory? Will the authors be collecting these from families, to examine adherence? This is unclear. 16. p. 10. No reference is given for the analytic methods described for the qualitative analysis. 17. p. 10. The authors describe for the first time the Family Consultation Group. What involvement will this group have in the feasibility study? 18. Figure 1. This figure – which is actually rather helpful and should be included in the main text – states that there will be follow-up assessments at 24 weeks post randomisation. This is not clear at all from the description provided on page 8. It is also unclear why the primary outcome variable is not being measured at this point. Please clarify.
--	--

REVIEWER	David Marshall Centre of Reviews and Dissemination University of York UK
REVIEW RETURNED	09-Aug-2018

GENERAL COMMENTS	The authors have made a good attempt at outlining a protocol for a study looking to examine the feasibility of running an RCT for a family therapy intervention on families of children with ASD. The impact on the family of a child with ASD is often overlooked in research so I believe a trial in this area would be of considerable value and due to the difficulties associated with recruitment for this area, a feasibility study is the right starting point. Overall, the manuscript is of good quality. It is well structured and reported, and includes all the elements to be expected in a feasibility study. I do have some minor concerns and suggestions for the authors’ consideration so I have listed these below. General  • The authors make liberal use of the term “maladaptive” throughout the document. This term has a very negative/medical quality which would be better to avoid if possible. The nice guidelines (NICE, no. 11) argues that it is also not an accurate term as research indicates these behaviours are quite adaptive and functional in some ways, and not disordered. The term
---

“challenging” behaviours should be used instead if possible or preferably “behaviour that challenges” if appropriate.

Introduction

- The authors have occasionally made unsubstantiated statements in the Introduction where references should be provided. These include the prevalence of ASD in the UK (page 1, line 30), research detailing the possibility of transgenerational element through insecure attachment problems (page 2, line 25) and a reference for SAFE page 2 line 54.
- I believe the prevalence is closer to 1.6% in the UK see Baron-Cohen et al., 2009; and Howlin & Moss, 2012.
- I have some concern about the authors focus on autism transgenerational transmission of ASD in the second paragraph. I don't believe there is sufficient evidence for this theory currently and it concerns me that it could be interpreted as symptoms of autism being “caused” by parental factors. I realise that this is not what the authors intended to relay but there isn't enough information presented for the reader to ascertain this. The readers should be aware of the potential implications of this line of research and highlight that autism is a neurodevelopmental disorder. If the author wishes to highlight how the findings of the transgenerational work may impact on SAFE or some of the symptoms of autism, it needs to be described in greater detail, critiqued and referenced with clear rationale for its inclusion.

Methods

- Inclusion criteria: Diagnosis, how will this diagnosis be checked and who will make it?
- Exclusion criteria: What is the cutoff for intellectual impairment and how will it be checked?
- Exclusion criteria: How will risk to staff be assessed?
- Study Design: No justification is given for the 2:1 ratio. Please describe why this is appropriate.
- Outcome Measures: I am not familiar with the SCORE measure but the authors have not backed up their claims that it is reliable and valid. Please provide references for these.

Ethics

- I believe the authors have underestimated the potential for adverse events and ethical impacts of the study. At the least, they should explore the ethical impact of disruption to the child's schedule, which is an important consideration for children with ASD. Additionally, due to the chosen recruitment method, some families may have to wait a considerable time for treatment due to waiting for a cohort to be formed. This should be justified.

Data protection

- What do the authors mean by pseudo-anonymity?

Research Governance

- This section does not seem to say how the study will be managed.

VERSION 1 – AUTHOR RESPONSE

Reviewer: 1

Reviewer Name

Liz Pellicano

Institution and Country

Macquarie University, Australia

Please state any competing interests or state 'None declared':

None declared.

Please leave your comments for the authors below

bmjopen-2018-025006

This manuscript reports a study protocol for a future, feasibility randomised controlled trial for a family-level therapeutic intervention for families of children on the autism spectrum. Many families struggle following the diagnosis of their child and so it is encouraging to see researchers focus their efforts on designing and evaluating post-diagnostic support for families.

I have several comments for the authors to consider – particularly with regard to clarifying various aspects of the rationale and methods. These are listed below. To begin, however, I note that the background section was overwhelmingly negative. The authors talk about the “substantial economic burden of autism”, the “palpable air of tension” in families, children’s “maladaptive behaviour and lack of empathy” to pick just a few phrases – all of which are potentially very stigmatising. This may be the reality for many families, but it is not a necessary consequence of having a child with a diagnosis of autism, with some studies describing families’ positive perceptions and experiences (see <https://www.tandfonline.com/doi/abs/10.1080/09687599.2016.1216393> and <https://www.ncbi.nlm.nih.gov/pmc/articles/PMC4827600/>, as two examples). Indeed, I assume that developing positive family wellbeing is the goal of this particular, SAFE, intervention. If that is the case – and this should be clearer in the introduction or in the description of the intervention itself – then the authors need to demonstrate that developing positive family wellbeing is possible for families. I suggest the authors rewrite their introduction to ensure that their description of family functioning is more balanced. The introduction has been re-written to give a more balanced view and to include the references suggested (pages 3-5).

I also found that there were a number of claims that were unsubstantiated (i.e., no references) and the rationale for various decisions were not provided. I have described these instances below, too.

1. p. 2, abstract. The authors state that children will be aged 3-6 years but this is not what is reported in the methods. Please correct. In fact, no rationale is given for the age range selected – which is a significant oversight, since age of diagnosis might well have an effect on the family’s response to therapy. The authors need to outline their rationale for the reader. The age range has been clarified and a rationale provided (page 5).

2. p. 3, para 1. Please provide a reference for the prevalence estimate. Please also provide a reference for the claim that “numbers are rising” in the UK. As far as I am aware, there is no prevalence study showing such a rise. In fact, the study of which I am aware suggests that – unlike in the US – there has been no change in prevalence, at least from 1990s through 2010: <https://bmjopen.bmj.com/content/3/10/e003219> the reference to rising numbers has been removed and a reference added for the prevalence estimate (page 3)

3. p. 4, para 1. The authors need to be more cautious with their claims about autism. For example, they state that autistic children demonstrate a “lack of empathy”, which is not necessarily the case. Children on the autism spectrum may well show difficulties in theory of mind and perspective-taking but do not necessarily show poor empathy. Please adjust this sentence. The sentence has been adjusted (page 4).

4. p. 4, para 1. No reference is provided for the final sentence on attachment in this paragraph – which sounds rather tenuous. The sentence has been removed (page 4)

5. p. 4 para 2. More details need to be provided regarding unpublished work, which has not been peer-reviewed and which cannot be examined by the reader. The reference to unpublished work has been replaced with references to two published articles (page 4)

6. p. 4, para 2. The authors state that “families of children with autism are often characterised by a palpable air of tension”. Is the phrase “palpable air of attention” from the paper cited, or is this the authors’ interpretation of their results? If it’s the former, the authors should provide a page number. If it’s the latter, I suggest that the authors cite qualitative studies that have formally analysed parents’ own perceptions of family functioning to support their claim. This phrase has been removed (page 4)

7. p. 5 para 2. The target population are families of children on the autism spectrum who do not have an intellectual disability. But no rationale is provided for excluding children with additional intellectual disabilities. And indeed, nothing is presented in the methods to show how they will ascertain the intellectual ability of the child – and therefore address their inclusion/exclusion criteria. A rationale is included explaining the exclusion of intellectual impairment (page 5) and how intellectual impairment will be ascertained (page 7).

8. p. 6 para 2. The authors state that they will be excluding children with a severity level of 3. Why exactly? Do they not believe that their intervention will be effective with this group? This needs to be properly explicated in the methods. This is explained (page 5)

9. p. 6, para 3. Similarly, no rationale is provided for decisions concerning the duration of the intervention and follow-up period. In particular, why is there no follow-up immediately following the intervention (16 weeks) or at the group follow up (24 weeks)? The 24 week follow-up is a group session where individual collection of data would not be feasible. This is explained in more detail (page 10)

10. p. 7, clinical outcome measures. Insufficient psychometric information was provided on all of these measures. Reliability estimates should be provided. It would also be helpful, particularly for the primary outcome measure, which will be less well known to readers, for the authors to provide some example items and describe the scale that parents will need to use in response to the items. Psychometric information has been provided for the proposed primary and secondary outcome measures along with associated references. The primary measure is freely available online and a link is provided in the references (pages 8-9).

11. p. 7. In the background provided, the authors suggest that the presence of the broader autism phenotype in parents may well moderate – and potentially mediate – family/children’s functioning. Yet I was surprised that there was no questionnaire measuring the BAP included in the secondary outcome measures. Perhaps this is another reason to revisit the background, as suggested above. The background has been revised

12. p. 7. Resource Use Questionnaire – is this measure meant to elicit information on the family’s ‘support as usual’? This is very unclear. Will it elicit information on the many types of interventions that parents might try to help their children (e.g. complementary and alternative medicine) beyond their accessing formal support services? The purpose of the Resource Use Questionnaire is provided in more detail (page 9)

13. p. 7. The qualitative methods should try to elicit information about potential harms of the intervention, as these could well be quite subtle in nature. The fact that the qualitative methods explore possible harm has been clarified (page 9)

14. p. 9. Please provide references for “the known visual processing preferences” of autistic people. References have been provided (page 9)

15. p. 9. Is the Between Session homework activity mandatory? Will the authors be collecting these from families, to examine adherence? This is unclear. The use of the BSA has been clarified including the fact that completion by participants will be recorded (page 10)

16. p. 10. No reference is given for the analytic methods described for the qualitative analysis. Two references have been added (page 12).

17. p. 10. The authors describe for the first time the Family Consultation Group. What involvement will this group have in the feasibility study? A section has been added on patient and public involvement (pages 12-13).

18. Figure 1. This figure – which is actually rather helpful and should be included in the main text – states that there will be follow-up assessments at 24 weeks post randomisation. This is not clear at all from the description provided on page 8. It is also unclear why the primary outcome variable is not being measured at this point. Please clarify. The figure is now included in the main text (page 11). The follow-up is a group session which is not appropriate for collecting data from individuals, this is clarified (page 10)

Reviewer: 2

Reviewer Name

David Marshall

Institution and Country

Centre of Reviews and Dissemination
University of York
UK

Please state any competing interests or state ‘None declared’:

None declared

Please leave your comments for the authors below

The authors have made a good attempt at outlining a protocol for a study looking to examine the feasibility of running an RCT for a family therapy intervention on families of children with ASD. The impact on the family of a child with ASD is often overlooked in research so I believe a trial in this area would be of considerable value and due to the difficulties associated with recruitment for this area, a feasibility study is the right starting point.

Overall, the manuscript is of good quality. It is well structured and reported, and includes all the elements to be expected in a feasibility study. I do have some minor concerns and suggestions for the authors’ consideration so I have listed these below.

General

- The authors make liberal use of the term “maladaptive” throughout the document. This term has a very negative/medical quality which would be better to avoid if possible. The nice guidelines (NICE,

no. 11) argues that it is also not an accurate term as research indicates these behaviours are quite adaptive and functional in some ways, and not disordered. The term “challenging” behaviours should be used instead if possible or preferably “behaviour that challenges” if appropriate. The term ‘maladaptive’ has been replaced with ‘challenging throughout.

Introduction

- The authors have occasionally made unsubstantiated statements in the Introduction where references should be provided. These include the prevalence of ASD in the UK (page 1, line 30), research detailing the possibility of transgenerational element through insecure attachment problems (page 2, line 25) and a reference for SAFE page 2 line 54. A reference has been added for prevalence and the introduction has been re-written to clarify or remove unsubstantiated claims (pages 3-5)
- I believe the prevalence is closer to 1.6% in the UK see Baron-Cohen et al., 2009; and Howlin & Moss, 2012. Since claims about prevalence vary we have stated that prevalence is ‘more than 1% (page 3)
- I have some concern about the authors focus on autism transgenerational transmission of ASD in the second paragraph. I don’t believe there is sufficient evidence for this theory currently and it concerns me that it could be interpreted as symptoms of autism being “caused” by parental factors. I realise that this is not what the authors intended to relay but there isn’t enough information presented for the reader to ascertain this. The readers should be aware of the potential implications of this line of research and highlight that autism is a neurodevelopmental disorder. If the author wishes to highlight how the findings of the transgenerational work may impact on SAFE or some of the symptoms of autism, it needs to be described in greater detail, critiqued and referenced with clear rationale for its inclusion. The clarity and balance of the introduction has been improved and unsubstantiated claims removed (pages 3-5).

Methods

- Inclusion criteria: Diagnosis, how will this diagnosis be checked and who will make it? Diagnosis will be made by clinical staff. This is clarified in the text (page 6)
- Exclusion criteria: What is the cutoff for intellectual impairment and how will it be checked? Criteria for identification of intellectual impairment are now listed (page 7)
- Exclusion criteria: How will risk to staff be assessed? This is now explained in the exclusion criteria (page 6).
- Study Design: No justification is given for the 2:1 ratio. Please describe why this is appropriate. Advantages and reasons for the 2:1 ratio are explained (page 7)
- Outcome Measures: I am not familiar with the SCORE measure but the authors have not backed up their claims that it is reliable and valid. Please provide references for these. Psychometric information regarding the primary measure and associated references are now included (page 8)

Ethics

- I believe the authors have underestimated the potential for adverse events and ethical impacts of the study. At the least, they should explore the ethical impact of disruption to the child’s schedule, which is an important consideration for children with ASD. Additionally, due to the chosen recruitment method, some families may have to wait a considerable time for treatment due to waiting for a cohort to be formed. This should be justified. A section on the procedure for serious adverse events has

been added (pages 13-14). Research staff will discuss estimated waiting times with families prior to consent. This has been clarified (page 15)

Data protection

- What do the authors mean by pseudo-anonymity? This is now explained (page 15).

Research Governance

- This section does not seem to say how the study will be managed. Further information about study management has been added (pages 15-16)

VERSION 2 – REVIEW

REVIEWER	Liz Pellicano Macquarie University, Australia
REVIEW RETURNED	05-Dec-2018

GENERAL COMMENTS	The authors have made some changes to their manuscript based on the reviewers' comments. But these changes were not as thorough as one might have hoped. I detail my continued concerns below.  1. The authors now state why they have included autistic children of severity levels 1 or 2 (and not 3) and report consistently that they have included children of a particular age range (3 – 16 years) throughout the manuscript and, they have nevertheless neglected to address my original concern regarding the age range. As far as I can tell, no rationale is given for the age range selected – which is especially important since (1) age of diagnosis might well have an effect on the family's response to therapy and (2) that the use of a play-based approach might not be appropriate for autistic teenagers (and actually could be quite patronising). The authors need to outline their rationale for the reader. 2. The authors now clearly state the ways in which they will gather information about IQ and what the IQ cut-off score will be for this study, they omit to tell the reader precisely *how* they will measure IQ (i.e., with what instrument). This is important since the authors should be well aware that severity levels do not correlate straightforwardly with IQ (e.g., cognitively-able or so-called 'high functioning' individuals do not necessarily function highly and can present with severity level 3), which means that some of the subjective criteria described may well exclude some of the very children that the authors might wish to include. This issue requires further clarification. 3. One issue that I had missed in my initial review was one of the qualitative outcomes and, in particular, participants' reasoning for declining and withdrawing from the study. Ethical guidelines usually state that participants are in no way obliged to provide a reason if they choose to withdraw from a study. How will the authors avoid coercion (to provide information when they might not wish to) in this instance?
---

	4. The authors state (p. 13) that “Families of children with autism are a vulnerable group”. Not all such families are vulnerable and labelling them as such is potentially disempowering. Please consider rewriting this statement to include that they are a “potentially vulnerable group”. 5. It was good to see more information on patient and public involvement, including the bringing together of a Family Consultation Group. It was unclear, however, what formal structures have been put in place to both ensure the involvement and to avoid tokenism. Please clarify whether you have set up regular meetings with this Consultation Group and how they will be involved in the research itself. Also, will they get paid for their involvement and experiential expertise (as they should)? This should be stated here. 6. The authors state the value of their Family Consultation Group, including in dissemination (see p. 12), but in the Dissemination section (p. 14), their input is notably absent. Please clarify. 7. Finally, I still felt that the broader autism phenotype featured heavily in the introduction, leading the reader to expect that the research will examine these traits in some way. Please consider revising this section again – especially given Reviewer 2’s concerns about people interpreting this section as the behaviours exhibited by autistic children in some way being ‘caused’ by parental factors, with which I very much agree.
--	--

REVIEWER	David Marshall Centre for Research and Dissemination UK
REVIEW RETURNED	05-Dec-2018

GENERAL COMMENTS	I believe the authors have adequately responded to my suggestions
---

VERSION 2 – AUTHOR RESPONSE

Reviewer(s)' Comments to Author:

Reviewer: 2

Reviewer Name: David Marshall

Institution and Country: Centre for Research and Dissemination

UK

Please state any competing interests or state ‘None declared’: None Declared

Please leave your comments for the authors below

I believe the authors have adequately responded to my suggestions. Thank you for your feedback

Reviewer: 1

Reviewer Name: Liz Pellicano

Institution and Country: Macquarie University, Australia

Please state any competing interests or state 'None declared': None declared

Please leave your comments for the authors below

The authors have made some changes to their manuscript based on the reviewers' comments. But these changes were not as thorough as one might have hoped. I detail my continued concerns below.

1. The authors now state why they have included autistic children of severity levels 1 or 2 (and not 3) and report consistently that they have included children of a particular age range (3 – 16 years) throughout the manuscript and, they have nevertheless neglected to address my original concern regarding the age range. As far as I can tell, no rationale is given for the age range selected – which is especially important since (1) age of diagnosis might well have an effect on the family's response to therapy and (2) that the use of a play-based approach might not be appropriate for autistic teenagers (and actually could be quite patronising). The authors need to outline their rationale for the reader. The available data from our recruiting centres suggests that the youngest age eligible children are currently diagnosed is 3 years. Seeking to recruit families of children with autism below this age would not be possible. In addition, one of our funders specified that the study should focus on 'children of school age'. Hence we chose 3-16 years. These reasons are given in the last paragraph of page 5. SAFE is best seen as a flexible toolkit which can be used for people of all ages. It is our belief that a playful approach is non-threatening and can be helpful for adults and children when applied by experienced therapists. We have extended our information about SAFE to explain this further on pages 5 and 10.

2. The authors now clearly state the ways in which they will gather information about IQ and what the IQ cut-off score will be for this study, they omit to tell the reader precisely *how* they will measure IQ (i.e., with what instrument). This is important since the authors should be well aware that severity levels do not correlate straightforwardly with IQ (e.g., cognitively-able or so-called 'high functioning' individuals do not necessarily function highly and can present with severity level 3), which means that some of the subjective criteria described may well exclude some of the very children that the authors might wish to include. This issue requires further clarification. We have clarified measures used and acknowledged the issues mentioned above (See track changes on pages 5, 6 and 7). We would also like to point out that this is a feasibility study and our ability to recruit eligible families and the experiences of the families we recruit may well lead to changes or refinements in the recruitment process.

3. One issue that I had missed in my initial review was one of the qualitative outcomes and, in particular, participants' reasoning for declining and withdrawing from the study. Ethical guidelines usually state that participants are in no way obliged to provide a reason if they choose to withdraw from a study. How will the authors avoid coercion (to provide information when they might not wish to) in this instance? Further details of the family focus groups and how arrangements have been made to ensure families do not feel coerced are included as track changes on pages 9 and 10.

4. The authors state (p. 13) that "Families of children with autism are a vulnerable group". Not all such families are vulnerable and labelling them as such is potentially disempowering. Please consider rewriting this statement to include that they are a "potentially vulnerable group". This has been revised as suggested

5. It was good to see more information on patient and public involvement, including the bringing together of a Family Consultation Group. It was unclear, however, what formal structures have been

put in place to both ensure the involvement and to avoid tokenism. Please clarify whether you have set up regular meetings with this Consultation Group and how they will be involved in the research itself. Also, will they get paid for their involvement and experiential expertise (as they should)? This should be stated here. Information regarding structures to ensure involvement and payment for patient advisors is now included on page 13.

6. The authors state the value of their Family Consultation Group, including in dissemination (see p. 12), but in the Dissemination section (p. 14), their input is notably absent. Please clarify. Information about the involvement of our Family Consultation Group in the dissemination process is now included in track changes on page 15

7. Finally, I still felt that the broader autism phenotype featured heavily in the introduction, leading the reader to expect that the research will examine these traits in some way. Please consider revising this section again – especially given Reviewer 2's concerns about people interpreting this section as the behaviours exhibited by autistic children in some way being 'caused' by parental factors, with which I very much agree. We acknowledge the reviewers concerns and parts of the introduction, which may be taken to suggest that parents 'cause' autism have been removed and parts of the introduction have also been re-written (See page 4). We respectfully point out, however, that the causes of autism are not fully understood and this is a topic of much debate, as are the reasons for high levels of psychological distress among family members of children with autism. It is our belief that any family where members are experiencing mental health problems and distress will deplete family resources and impede positive family dynamics and it is this problem that SAFE seeks to address. It may be that the introduction has led the reviewer to misunderstand the focus of the intervention, which is not to address or explain the symptoms of autism within the child, but to support the whole family to build on strengths in order to face the challenges life throws at them. We believe the changes made demonstrate more clearly that the introduction aims to document the high risk of encountering psychological distress among family members of children with autism (and hence the need for SAFE).